# Molecular Characterization and Expression Patterns of Two Pheromone-Binding Proteins from the Diurnal Moth *Phauda flammans* (Walker) (Lepidoptera: Zygaenoidea: Phaudidae)

**DOI:** 10.3390/ijms24010385

**Published:** 2022-12-26

**Authors:** Lian Chen, Zhong Tian, Jin Hu, Xiao-Yun Wang, Man-Qun Wang, Wen Lu, Xiao-Ping Wang, Xia-Lin Zheng

**Affiliations:** 1Guangxi Key Laboratory of Agric-Environment and Agric-Products Safety, National Demonstration Center for Experimental Plant Science Education, College of Agriculture, Guangxi University, Nanning 530004, China; 2Xianning Academy of Agricultural Sciences, Xianning 437000, China; 3Hubei Key Laboratory of Insect Resources Utilization and Sustainable Pest Management, College of Plant Science and Technology, Huazhong Agricultural University, Wuhan 430070, China

**Keywords:** zygaenoidea, diurnal moth, pheromone-binding proteins, PflaPBP1, PflaPBP2, sex pheromone perception

## Abstract

Sex pheromone-binding proteins (PBPs) play an important role in sex pheromone recognition in Lepidoptera. However, the mechanisms of chemical communication mediating the response to sex pheromones remain unclear in the diurnal moths of the superfamily Zygaenoidea. In this study, *Phauda flammans* (Walker) (Lepidoptera: Zygaenoidea: Phaudidae) was used as a model insect to explore the molecular mechanism of sex pheromone perception in the superfamily Zygaenoidea. Two novel pheromone-binding proteins (*PflaPBP1* and *PflaPBP2*) from *P. flammans* were identified. The two pheromone-binding proteins were predominantly expressed in the antennae of *P. flammans* male and female moths, in which PflaPBP1 had stronger binding affinity to the female sex pheromones *Z*-9-hexadecenal and (*Z*, *Z*, *Z*)-9, 12, 15-octadecatrienal, PflaPBP2 had stronger binding affinity only for (*Z*, *Z*, *Z*)-9, 12, 15-octadecatrienal, and no apparent binding affinity to *Z*-9-hexadecenal. The molecular docking results indicated that Ile 170 and Leu 169 are predicted to be important in the binding of the sex pheromone to PflaPBP1 and PflaPBP2. We concluded that PflaPBP1 and PflaPBP2 may be responsible for the recognition of two sex pheromone components and may function differently in female and male *P. flammans*. These results provide a foundation for the development of pest control by exploring sex pheromone blocking agents and the application of sex pheromones and their analogs for insect pests in the superfamily Zygaenoidea.

## 1. Introduction

Moths detect pheromones with specialized chemosensilla localized on the antennae or, more rarely, on other parts of the body [1]. Normally, moth antennae can accurately identify and distinguish sex pheromones and their analogs by pheromone-binding proteins (PBPs) [2]. PBPs are either specific to or highly enriched in the antennae of males and act as pheromone carriers by binding and transporting odorant molecules across the antennal hemolymph to odorant receptor proteins [3,4]. Some studies have shown that the sexually dimorphic antennal flagellum of the male is enlarged so that the flagella can house extra arrays of pheromone-specific long trichoid sensilla, where PBPs have high binding affinities to sex pheromones released by female moths [5,6]. Further studies indicated that the female moth antennae can also detect their conspecific sex pheromones [7], and the expression of PBPs has been observed in antennae of female moths such as *Bombyx mori* (Linnaeus) (Lepidoptera: Bombycidae), *Antheraea polyphemus* (Cramer) (Lepidoptera: Saturniidae), and *Manduca sexta* (Linnaeus) (Lepidoptera: Sphingidae) [8,9,10]. Previous studies have suggested that female expression of PBPs was associated with a small number of specific olfactory sensilla on the antennae [11,12].

The earliest studies on PBPs were conducted in moths. In 1981, the first PBP (*ApolPBP*) was isolated from the antennae of male *A. polyphemus* [13]. Since then, many lepidopteran PBPs have been identified from *M. sexta* [14], *Lymantria dispar* (Linnaeus) (Lepidoptera: Erebidae) [15], *Spodoptera exigua* (Hübner) (Lepidoptera: Noctuidae) [16], *Agrotis ipsilon* (Hufnagel) (Lepidoptera: Noctuidae) [6], *Chilo suppressalis* (Walker) (Lepidoptera: Crambidae) [17], *Tryporyza intacta* (Snellen) (Lepidoptera: Pyralidae) [4], and *Carposina sasakii* Matsumura (Lepidoptera: Carposinidae) [18]. Many studies have indicated that PBPs from the antennae of male moths, such as *Plutella xylostella* (Linnaeus) (Lepidoptera: Plutellidae) [19], *C. suppressalis* [17], and *Helicoverpa armigera* (Hübner) (Lepidoptera: Noctuidae) [20], not only robustly bound the sex pheromone components but also significantly bound pheromone analogs. In contrast, extensive evidence has indicated that PBPs in female antennae facilitate the detection of sex pheromone components released by themselves or by conspecifics [21,22]. Interestingly, PBPs in females not only have a high binding affinity for sex pheromones but can also bind host-related semiochemicals, e.g., *Maruca vitrata* (Fabricius) (Lepidoptera: Crambidae) [23] and *L. dispar* [24].

Many studies have revealed the olfactory molecular mechanism of diurnal moths for discriminating sex pheromone components, for example, those in *L. dispar* [24], *H. armigera* [25], and *B. mori* [9,26]. A notable exception is represented however by castniid moths, which did not produce any pheromone to attract males, and mate location was achieved only visually by patrolling males [27]. Most species in the superfamily Zygaenoidea are diurnal moths that use double strategies (e.g., olfactory and visual signals) to identify male and female sexes [28,29]. Numerous studies on sex pheromones in the superfamily Zygaenoidea have focused mainly on the identification and field testing of the families Limacodidae [30,31], Phaudidae [32], and Zygaenidae [33,34,35,36,37,38,39] since the 1980s. However, the mechanism of chemical communication involved in the response to sex pheromones is still not well-known in diurnal moths of the superfamily Zygaenoidea.

*Phauda flammans* (Walker) (Lepidoptera: Zygaenoidea: Phaudidae) is a notorious leaf-eating pest on *Ficus* spp. in many countries in Southeast Asia and southern China [40,41,42,43,44,45]. *P. flammans* produces 2–3 generations per year in southern China [43]. This species is a monandrous moth, and its adults live for 4–5 days [29]. *P. flammans* adults are active during the photophase that lasts from 6:00 to 16:00 h and mate from 9:00 to 19:00 h [32,43]. Although different management strategies have been explored to control *P. flammans* [46,47], chemical control is still one of the main management techniques. However, the overuse of insecticides could result in insect resistance to the most commonly used products as well as environmental contamination [48]. Sex pheromone-based strategies (e.g., mass trapping and mating disruption) have been identified as effective methods of biological control for managing insecticide resistance and environmental pollution [7,49]. Our earlier study showed that the primary components of *P. flammans* female-produced sex pheromones were *Z*-9-hexadecenal and (*Z*, *Z*, *Z*)-9, 12, 15-octadecatrienal, and the capture of only males in field tests using the two synthetic pheromone components at a ratio of 1:1 confirmed the activity of the identified sex pheromone candidates [32]. Furthermore, sexual dimorphism in the antennal sensilla of *P. flammans* was observed, with the male antennae having more sensilla trichodea than the female antennae [50]. Unfortunately, the inherent relationship between the number or distribution of sensilla trichodea and sensitivity to female sex pheromones in *P. flammans* is still unclear, although the responses of sensilla trichodea to female sex pheromones in Lepidoptera are well-known [51,52,53,54]. Thus far, limited information is available about the molecular mechanism of sex pheromone perception in *P. flammans*.

In this study, *P. flammans* was used as a model insect to explore the molecular mechanism of sex pheromone perception of diurnal moths in the superfamily Zygaenoidea. Our objectives were to characterize the nucleotide and amino acid features, tissue-specific expression, binding affinity for sex pheromones, and candidate binding site via three-dimensional structure construction and molecular docking of *P. flammans* PBPs. These results will help elucidate the role of *P. flammans* PBPs in sex pheromone perception and may also suggest focused strategies to disrupt semiochemical detection and recognition for this moth in the field.

## 2. Results

### 2.1. Cloning and Sequence Analysis of PflaPBPs

Two PBP genes were successfully cloned using the cDNA of *P. flammans* and named *PflaPBP1* and *PflaPBP2*, which were submitted to the National Center for Biotechnology Information (NCBI) database and assigned accession numbers MK948015 and MK948014, respectively (Figure A1). The complete open reading frames (ORFs) of *PflaPBP1* and *PflaPBP2* are 495 bp and 492 bp, respectively. PflaPBP1 encodes 164 amino acids, while the signal peptide contains 22 amino acids. After removal of the signal peptide, the protein size was 15.84 kDa, and the isoelectric point was 4.89. PflaPBP2 encodes 163 amino acids, while the signal peptide contains 21 amino acids. After removal of the signal peptide, the protein size was 16.08 kDa, and the isoelectric point was 5.60.

The derived amino acid sequence was used to construct a phylogenetic tree with the other published insect PBPs (Figure 1). The PBPs of Lepidoptera were found to be very conservative. PfalPBP1 clustered in a group with other lepidopteran PBP1s, while PflaPBP2 clustered in a group with PBP2. PfalPBP1 and PflaPBP2 showed the highest homology with the moths of Pyralidae (e.g., DabiPBP of *Dioryctria abietella*, OnubPBP1 of *Ostrinia nubilalis* (Hübner) and LstiPBP of *Loxostege sticticalis* Linnaeus) and butterflies (e.g., BanyPBP of *Bicyclus anynana* and PrapPBP2 of *Pieris rapae* (Linnaeus)), respectively. Similar to other lepidopteran PBPs, both PflaPBP1 and PflaPBP2 have six typical cysteine residues (Figure A2).

### 2.2. Tissue Expression of PflaPBPs

The expression levels of *PflaPBP1* and *PflaPBP2* varied significantly among adult tissues (Figure 2, Table 1 and Table 2). Specifically, expression levels of *PflaPBP1* and *PflaPBP2* in the antennae were 241.17 times and 267.38 times the sum of other tissues, respectively. Differences in *PflaPBP1* expression were also found between sexes (Table 1). There were significant interactions between sex and tissues, showing that *PflaPBP1* and *PflaPBP2* are expressed at different levels in the same tissue in different sexes (Table 1 and Table 2). In particular, the expression of *PflaPBP1* was 1.56 times higher in the antennae of males than in those of females, and the expression of *PflaPBP2* was 1.57 times higher in the antennae of females than in those of males (Figure 2).

### 2.3. Prokaryotic Expression and Purification of PflaPBPs

Recombinant proteins were induced and expressed in *E. coli* prokaryotic cells in vitro. After induction by isopropyl-β-D-1-thiogalactopyranoside (IPTG), there were obvious protein bands between 25 and 35 kDa in the experimental group compared with the control group, indicating that the protein was successfully induced. Then, the histidine (His) tag was removed and purified to obtain the target soluble protein from the supernatant (Figure 3). The yield of purified protein was 0.21 mg/mL for PflaPBP1 and 0.07 mg/mL for PflaPBP2.

### 2.4. Test of Fluorescent Probe

To detect the binding affinity of proteins to small molecules by fluorescence competitive binding, N-phenyl-1-naphthylamine (1-NPN) was used as the probe to test the suitability of PflaPBP1 and PflaPBP2 at pH = 7.4. The maximum fluorescence value of the complex of the two proteins and 1-NPN increased gradually with the increase in the concentration of 1-NPN until it reached saturation. Scatchard plots show that 1-NPN binds the two proteins one-to-one under neutral conditions (Figure 4). Thus, 1-NPN can be used as a probe for fluorescence competitive binding experiments of PflaPBP1 and PflaPBP2.

### 2.5. Fluorescence Binding Affinities

The binding characteristics of PflaPBP1 and PflaPBP2 with two sex pheromones were analyzed by fluorescence competitive binding experiments (Figure 5). These results showed that the binding affinity of PflaPBP1 to ligands was bound with a higher affinity than the binding affinity of PflaPBP2 to ligands. PflaPBP1 had a moderate binding affinity for the female sex pheromones *Z*-9-hexadecenal (Ki = 21.26 μM) and (*Z*, *Z*, *Z*)-9, 12, 15-octadecatrienal (Ki = 38.48 μM), and PflaPBP2 had a moderate binding affinity for (*Z*, *Z*, *Z*)-9, 12, 15-octadecatrienal (Ki = 33.95 μM) but no detectable binding affinity for *Z*-9-hexadecenal (Ki = 108.68 μM) (Table 3).

### 2.6. Three-Dimensional Structure and Molecular Docking of PflaPBPs

The protein structure of BmorPBP1 was used as a template to construct the protein structures of PflaPBP1 and PflaPBP2. The consistency between PflaPBP1 and PflaPBP2 with the best template was different, with 67.38% (the evaluation of Ramachandran plots was 92.67%, Figure A3) and 47.89% (the evaluation of Ramachandran plots was 95.86%, Figure A4) sequence consistency, respectively. The three-dimensional structure consists of six α-helixes and one additional α-helix (α7-helix) (Figure 6A,B). In addition, three disulfide bonds are formed by the six conserved cysteine (Cys) residues to connect, which are α1-α3 (Cys45-Cys84), α3-α7 (Cys78-Cys142), and α6-α7 (Cys97-Cys117) in PflaPBP1 and α1-α3 (Cys50-Cys87), α3-α7 (Cys83-Cys143), and α6-α7 (Cys87-Cys152) in PflaPBP2 (Figure 6C). The Cys sites of PflaPBP1 and PflaPBP2 are very similar to the disulfide bond.

To explore the binding site of PflaPBP1 and PflaPBP2 with sex pheromones, we selected sex pheromones [*Z*-9-hexadecenal and (*Z*, *Z*, *Z*)-9, 12, 15-octadecatrienal] that possibly bind to PflaPBP1 and PflaPBP2 for three-dimensional structural docking. The results of molecular docking indicated that hydrogen bonding was the main interaction mode between PflaPBP1/PflaPBP2 and the tested sex pheromone molecules. *Z*-9-hexadecenal and (*Z*, *Z*, *Z*)-9, 12, 15-octadecatrienal showed hydrogen bonding to PflaPBP1. The hydrogen bonding sites of *Z*-9-hexadecenal are Leu 169 and Ile 170 (Figure 7A–C), while (*Z*, *Z*, *Z*)-9, 12, 15-octadecatrienal only bound to Ile 170 (Figure 7D–F); (*Z*, *Z*, *Z*)-9, 12, 15-octadecaenal showed hydrogen bonding to PflaPBP2, and the binding site was Lys 69 (Figure 7G–I). This hydrogen bond acts as a fixator to bind ligands at the binding site. *Z*-9-hexadecenal was bound to PflaPBP1 through two hydrogen bonds, indicating strong binding.

## 3. Discussion

Many species of Zygaenoidea release sex pheromones which can be used in the monitoring and forecasting of these diurnal moths in the field [31,34,35,36,37,38,39]. *P. flammans* sex pheromones have been used as biological control agents in the field, and a mixture of the two synthetic pheromones of *P. flammans*, *Z*-9-hexadecenal and (*Z*, *Z*, *Z*)-9, 12, 15-octadecatrienal, at a ratio of 1:1 could effectively attract male moths [32]. Since field trials are normally more convincing to identify sex pheromones in insects, *Z*-9-hexadecenal and (*Z*, *Z*, *Z*)-9, 12, 15-octadecatrienal were proven to be female-produced sex pheromones that could attract males [32]. Based on our findings, we deduced that PflaPBP1, but not PflaPBP2, was the dominant binding protein in *P. flammans* as a passive carrier delivering sex pheromones to neuronal membranes. First, a mixture of *Z*-9-hexadecenal and (*Z*, *Z*, *Z*)-9, 12, 15-octadecatrienal at a ratio of 1:1 was effective in attracting male moths [32], which indicated that the recognition of both chemicals was needed. Second, PflaPBP1 was in accordance with the male-biased expression in antennae (Figure 2A). In addition, PflaPBP1 has binding affinity to both sex pheromones, while PflaPBP2 could only bind *Z*-9-hexadecenal (Figure 5).

Our results also indicated that *PflaPBP1* and *PflaPBP2* may function differently in females and males. Quantitative real-time PCR analysis of the *PflaPBP1* and *PflaPBP2* genes showed that they were predominantly expressed in the antennae of both sexes in *P. flammans*, similar to the findings in several diurnal moths, e.g., *M. sexta* [8], *A. polyphemus* [10], and *B. mori* [26]. Interestingly, the transcript levels of the *PflaPBP* genes in the antennae of both sexes were different. For example, the expression level of *PflaPBP1* in male antennae was significantly higher than the expression level of *PflaPBP1* in female antennae and vice versa for *PflaPBP2* (Figure 2). Further analysis of the binding characteristics of *PflaPBP1* with the two sex pheromones indicated an important role in female sex pheromone perception. Therefore, the male-biased expression level of *PflaPBP1* suggested that the gene may be involved in sexual communication involving pheromones. For the *PflaPBP2* gene, the expression level in female antennae was significantly higher than the expression level in male antennae, and the protein could bind (*Z*, *Z*, *Z*)-9, 12, 15-octadecatrienal. In this context, the presence of the *PflaPBP2* gene in the olfactory sensilla of female antennae suggests that *P. flammans* females may detect either their own sex pheromone or at least certain components of their sex pheromone. There are two hypotheses to explain these results. First, females detecting sex pheromones reproduced by themselves are believed to allow the detection of competing plumes for avoidance [17,55]. Especially for the monandrous moth *P. flammans*, conspecific signals are used as cues for rapid relocation to valuable sites to avoid competition in their short lifespan because delayed mating significantly reduces their reproductive outputs [56]. Thus, conspecific communication with pheromones may improve the chances of mating success [57,58]. Second, female expression of *PflaPBP2* may help females recognize the *Z*-9-hexadecenal and (*Z*, *Z*, *Z*)-9, 12, 15-octadecatrienal released by other females, which is used to avoid the identification bias of males among similar species, such as *P. triadum* Walker (Lepidoptera: Phaudidae), leading to cross-mating of different species [7]. Indeed, female autodetection of their sex pheromones has been demonstrated for several diurnal moths, including *Panaxia quadripunctaria* Poda (Lepidoptera: Arctiidae) [59] and *Utetheisa ornatrix* (Linnaeus) (Lepidoptera: Erebidae) [60]. Our results suggest that *PflaPBP1* and *PflaPBP2* may play different roles in female sex pheromone perception and that these genes may undergo functional differentiation.

In this study, sex pheromone components of *P. flammans*, *Z*-9-hexadecenal and (*Z*, *Z*, *Z*)-9, 12, 15-octadecatrienal, were bound with PflaPBP1 and PflaPBP2 with different degrees of sensitivity. The three-dimensional structure consists of six α-helixes and one additional α-helix (α7-helix) (Figure 6A,B), which may be involved in the binding and release of pheromones under different conditions. For example, PflaPBP1 showed binding to the female sex pheromones *Z*-9-hexadecenal and (*Z*, *Z*, *Z*)-9, 12, 15-octadecatrienal, while PflaPBP2 showed binding only to (*Z*, *Z*, *Z*)-9, 12, 15-octadecatrienal (Figure 5). Both sex pheromone ligands showed the same chain length based on the three-dimensional structure models. One end of each molecule had hydroxyl groups, which tended to form hydrogen bonds with PBP residues. These hydrogen bonds were considered to be the main interaction mode between PflaPBPs proteins and sex pheromone ligands, which has been confirmed in several insect PBPs [61,62]. Furthermore, the docking results showed that both the binding sites of PflaPBP1 and the two sex pheromone ligands involved Ile 170. We speculated that Ile 170 might be the key site for the recognition of PflaPBP1.

The preceding studies found that the binding affinity of PBPs for pheromones could be significantly decreased or totally discontinued even with the mutation of one or two amino acids [63,64]. A study also conjectured that some hydrophobic residues and their hydrophobic interactions were essential for binding to their sex pheromone components [65]. Moreover, further functional verification of *PflaPBPs* by RNAi survey rather than point mutation should provide direct biological evidence that was fully considered. Unfortunately, PflaPBP1 and PflaPBP2 could not be effectively depressed by RNA interference after many attempts (unpublished data). We deduced that low stability and intracellular transport of dsRNA might lead to a poor RNAi response in *P. flammans*, which was consistent with other lepidopterans such as *Spodoptera littoralis* (Boisduval) and *Chrysodeixis includens* (Walker) [66,67,68,69,70,71,72]. Furthermore, the clustered regularly interspersed short palindromic repeats (CRISPR)/Cas9 technique as a substitutive method for RNAi would be certain to facilitate functional research of PflaPBP1 and PflaPBP2, combined with electrophysiological and behavioral assays to verify their functions [65], which cannot be implemented at present because we were unable to collect *P. flammans* eggs successfully due to the lower mate success in the laboratory. For this reason, we have taken transgenerational breeding in *P. flammans* as a priority for the prospective study of gene function via the CRISPR/Cas9 technique, which is currently achieving limited progress.

There is a possibility that other sex pheromone components participate in sex pheromone recognition in *P. flammans*, which must be considered. First, we note that the identification of sex pheromone components and their proportion is a process of constant and precise adjustment in many species of lepidoptera, such as *Spodoptera frugiperda* (J. E. Smith) [73]. Meanwhile, there is also the possibility of a novel sex pheromone component in *P. flammans* that might be identified in the future with new technology development. Thus far, *Z*-9-hexadecenal and (*Z*, *Z*, *Z*)-9, 12, 15-octadecatrienal have been proved as two vital components by a series of laboratory and field studies [32]. In this context, we focus on the known sex pheromone components and their interactions with PflaPBP1 and PflaPBP2.

Only two *PflaPBPs* were involved in this study, while 3~4 PBPs have been identified in some lepidopteran species [74,75]. These two *PflaPBPs* were originally identified from an antenna Illumina high-throughput transcriptome (unpublished data) in *P. flammans*, and their binding affinity for *Z*-9-hexadecenal and (*Z*, *Z*, *Z*)-9, 12, 15-octadecatrienal was also confirmed. However, we noticed that there might be more than two *PflaPBPs* in *P. flammans* just like *B. mori* [76]. First, a more advanced antenna transcriptome via PacBio full-length sequencing was established to find a third *PflaPBP* with no success [77]. Second, local basic local alignment search tool (BLAST) was also conducted, and no additional *PflaPBP* was found. These attempts convince us that it is very likely to have only two *PflaPBPs* in *P. flammans*. Moreover, the number of PBPs may be caused by species specificity because only two PBPs have also been reported in many insects, such as *Sitotroga cerealella* (Olivier) [78]. However, 58 OBPs were identified in *P. flammans* [77]. There is a possibility that some of them might act as a PBP in addition to *PflaPBP1* and *PflaPBP2* with a grain of salt because their functional studies have not yet been conducted.

Currently, studies on sex pheromone binding proteins of Lepidoptera are focused mainly on nocturnal moths, while similar studies on diurnal moths remain unclear. For the first time, we illustrated the interactions between the known sex pheromones and the identified pheromone binding proteins in vitro, which have a guiding role in probing the olfactory molecular mechanism of sex pheromone perception in the superfamily Zygaenoidea as well as other diurnal species.

In conclusion, PflaPBP1 and PflaPBP2 are responsible for the recognition of *Z*-9-hexadecenal and (*Z*, *Z*, *Z*)-9, 12, 15-octadecatrienal to varying degrees. These findings may facilitate pest control by exploring sex pheromone blocking agents. Our research will also help with the application of sex pheromones and their analogs in the diurnal moths of superfamily Zygaenoidea.

## 4. Materials and Methods

### 4.1. Insect Rearing and Sample Collection

From June to July 2018, *P. flammans* larvae were collected on *Ficus benjamina* L. trees located at the campus of Guangxi University (Nanning City, Guangxi Zhuang Autonomous Region, China; 108.17° E, 22.50° N). The collected larvae were then routinely placed into plastic boxes (radius = 8.0 cm, height = 12 cm; 10 individuals/box) and incubated in a standardized insect rearing room. The room was set to 26 (±1) °C, with a relative humidity of 60–80% and a photoperiod of L 14: D 10. Fresh young leaves of *F. benjamina* were replaced daily to feed the larvae until the pupae were collected. The pupae were then sexed following Mao et al. [79] and kept individually in plastic tubes (2.7 cm diameter, 10.3 cm height) with a plastic lid with 10 holes (2 mm diameter) in the middle for ventilation. Emerged adult moths were reared with 10% sucrose solution before use.

The antennae, heads (without antennae), thoraxes, abdomens, legs, and wings were dissected from 10 living individuals by surgical scissors (Shandong Xinhua Medical Devices Co., Ltd., Zibo, China) to prepare tissue samples. The sampled tissues were frozen immediately in liquid nitrogen after collection and stored at −80 °C. Three biological repeats were prepared.

### 4.2. RNA Extraction and Cloning of PflaPBPs

Total RNA was extracted from samples of *P. flammans* using TRIzol reagent (Takara, Beijing, China). Total RNA was examined by a NanoDrop 2000 system (Thermo Scientific™, Waltham, MA, USA) and 1% agarose gel electrophoresis (Newbio, San Francisco, CA, USA). Total RNA with a clear band and an OD260/280 value of 1.9–2.1 was used for further cDNA preparation. cDNA preparation was performed in accordance with the instructions of the PrimeScript RT reagent Kit with gDNA Eraser (Perfect Real Time) (Takara, Beijing, China, RR047A) and stored at −20 °C.

PBP and pheromone-binding protein were used as keywords to screen the annotated chemosensory genes by the results of Nr annotation from the antenna transcriptome database. So, candidate PBPs were identified from the antenna transcriptome database of *P. flammans* (unpublished data) and applied to primer design for the ORF (Table 4) by PCR with gene-specific primers. The total PCR mixture of 20.0 µL contained 14.6 µL of double-distilled (dd) H_2_O, 1.6 µL of dNTPs (2.5 mM, TransGen Biotech, Beijing, China), 2.0 µL of HiFi Buffer II, 0.2 µL of HiFi (TransTaq DNA Polymerase High Fidelity, TransGen Biotech, Beijing, China), 0.4 µL of forward primer (10 µM), 0.4 µL of reverse primer (10 µM), and 0.8 µL of sample cDNA. The PCR cycling conditions were as follows: initial denaturation at 95 °C for 1 min; then, 33 cycles of 95 °C for 30 s, 60 °C for 30 s, and 72 °C for 20 s; and a final extension at 72 °C for 5 min. ORF cloning was performed by PCR using polymerase. Then, the PCR product was linked to the pMD18-T vector (Takara, China) and transformed into Trans-t1 competent cells (preparation by us) according to the product manual. Sequence comparison of the cloned genes was performed.

### 4.3. Sequence Analysis

The homology was first compared by the BLAST of the NCBI database, and then, the amino acid sequence of the obtained sequence was predicted and deduced by using ExPASy online protein analysis website (https://web.expasy.org/translate/, accessed on 3 November 2022). Genedoc version 2.0 software was used for protein sequence alignment. MEGA 7.0 software was subsequently used to construct the phylogenetic tree of PflaPBPs and other lepidopteran PBPs [80].

### 4.4. Tissue Expression of PflaPBPs

The expression levels of *PflaPBPs* in different tissues of adult moths were measured by qRT-PCR. *GAPDH* and *TUB2* were used as internal reference genes [81]. Each treatment had three biological repeats and three technical repeats. The related qRT-PCR primers are listed in Table 3. Primers for qRT-PCR were evaluated before the expression evaluation with five times gradient dilution of the cDNA as DNA template to calculate PCR efficiency(E): E = 10^−1/k^ − 1, where k is the slope of standard curve and regression coefficient: Cq = −klgX_0_ + b, where Cq is the threshold cycle, k is the slope of standard curve, X_0_ is the template concentration. The PCR system (10 µL) was prepared on ice according to the following system: 5 µL of SYBR^®^ Premix Ex Taq™ II (Tli RNaseH Plus), 2.6 µL of ddH_2_O, 0.2 µL of each primer (forward and reverse primers, 10 µM), and 2.0 µL of cDNA template. The qRT-PCR procedure was as follows: 95 °C for 3 s; 40 cycles at 95 °C for 5 s and at 60 °C for 30 s; and melting curve analysis at 95 °C for 15 s; 60 °C for 1 min; and 95 °C for 15 s. For tissue expression of *PflaPBPs*, cDNA samples were all diluted 20 times, and the obtained data were normalized by the 2^−ΔΔCt^ method [82].

### 4.5. Expression and Purification

The target fragments PflaPBP1 and PflaPBP2 were subcloned into the *BamHI* and *XhoI* sites of pATX-sumo expression vectors. Single colonies containing recombinant plasmids were selected from the plate and inoculated in 5 mL of Luria-Bertani (LB) medium containing kanamycin (50 μg/mL) overnight at 37 °C. Subsequently, 200 μL was inoculated into 20 mL of LB medium containing antibiotics and incubated at 37 °C until OD_600_ = 0.6. Then, the cells were transformed into T7E and BL21 strains, and isopropyl-β-D-thiogalactopyranoside (IPTG) was added at 16 °C for 16 h. The target protein with an N-terminal 6*His tag was expressed and purified. Samples were collected, and 12,000 rpm centrifugation for 1 min was used to collect the bacteria. An ultrasound device was used to lyse bacterial cells suspended in 200 μL of phosphate-buffered saline (PBS). The ultrasound time was 2 s, and the interval was 2 s, totaling 4 min. The supernatant and precipitate were separated by centrifugation at 12,000 rpm for 2 min. The retained final PflaPBPs were in the supernatant and applied for purification by nickel column affinity chromatography (Thermo, USA) according to the manufacturer’s specifications with two washes. Recombinant bovine enterokinase was added to the eluted proteins and incubated at 26 °C for 10 h to remove the His-tags from the recombinant proteins. Sodium dodecyl sulfate-polyacrylamide gel electrophoresis (SDS-PAGE) (12%, 80 V concentrated gel, 20 min, 120 V separated gel, 45 min) was performed to assess the protein expression and purification. The purified protein was dialyzed in Tris buffer (pH 7.4 and pH 5.0), subpackaged and frozen in a −80 °C ultralow temperature refrigerator.

### 4.6. Probe Test and Fluorescence Binding Assays

An RF-5301PC fluorescence spectrophotometer (Shimadzu, Kyoto, Japan) was used in this experiment for fluorescence competitive binding analysis with a 1-cm quartz colorimetric cup at room temperature. N-Phenyl-1-naphthylamine (1-NPN) was used as the probe and applied for the suitability test. The 1-NPN was excited at 337 nm, in which the emission spectra were recorded between 350 and 660 nm. The purified PflaPBP1 and PflaPBP2 were diluted to a final concentration of 2 µM of the protein solution in Tris-HCl buffer and titrated with aliquots of 1 mM 1-NPN dissolved in methanol, which were 0, 2, 4, 8, 12, 16, 20 µM. The maximum fluorescence value (manipulate peak pick) of each excitation was recorded. The binding data were obtained from three independent measurements. The dissociation constants of PflaPBP1/PflaPBP2 for 1-NPN were calculated from Scatchard plots of the binding data by using Prism 5 software (GraphPad, La Jolla, CA, USA) with nonlinear regression for the one site-specific binding method.

The two sex pheromones *Z*-9-hexadecenal (Z9H, 96.3% chemical purity) and (*Z*, *Z*, *Z*)-9,12,15-octadecatrienal (Z9O, 93% chemical purity) were obtained from Shanghai T & W Pharmaceutical Co., Ltd. (Shanghai, China) and Pherobank BV (Wijk bij Duurstede, The Netherlands) [32]. The competitive binding of these two sex pheromone components was measured using 1-NPN (2 µM) as the fluorescent probe with a stoichiometry of 1:1 (protein: ligand). PflaPBP1 or PflaPBP2 was titrated to a final concentration of 2 μmol/L with 50 mM Tris-HCl buffer. The sex pheromone dissolved in methanol was gradually added to the mixed solution of protein and 1-NPN with the final concentration of each ligand added to the sample ranging from 0 μM to 20 μM. All sets of fluorescence spectrophotometers were the same as above. The binding data were obtained from three independent measurements. Reduction in relative fluorescence intensity indicated that the competing compound displaced 1-NPN from the binding site of PflaPBP1 or PflaPBP2. The following formula was adopted to calculate the binding affinities (*Ki*): $Ki=\frac{\left[IC_{50}\right]}{1+\left[1-NPN/K_{1-NPN}\right]}$) where [1-NPN] is the free concentration of 1-NPN, [*IC50*] is the ligand concentration replacing 50% of the fluorescence reporter, and K_1-NPN_ is the dissociation constant of the PflaPBP/1-NPN complex. The data were fitted with nonlinear regression model using Prism software (version 5.0).

### 4.7. Three-Dimensional Structures and Molecular Docking of PflaPBPs

With the SWISS MODEL website (https://swissmodel.expasy.org/interactive, accessed on 3 November 2022), PflaPBP1 and PflaPBP2 were compared to proteins in the CSBP Protein Data Bank (PDB). The selected template sequence must be more than 30% similar to the target sequence [83]. As a model insect, *B. mori* is the most advanced sex pheromone binding protein in the lepidopteran family. The similarity of BmorPBP (PDB ID: 1GM0.1) to PflaPBP1 and PflaPBP2 was 57.23% and 38.15%, respectively. Therefore, this BmorPBP was selected as the structural template to establish the three-dimensional structure model of PflaPBP1 and PflaPBP2.

The protein was pretreated by AutoTools and then calculated by AutoGrid. The active sites were defined as potential sites calculated by POCASA (http://altair.sci.hokudai.ac.jp/g6/service/pocasa/, accessed on 3 November 2022), and grid box coordinates were set. PflaPBP1 is (34.946, −11.031, 38.975), and the box size is 20 × 20 × 20 mesh points; PflaPBP2 is (9.028, 4.576, 32.343), the box size is 20 × 20 × 20 mesh points, and the distance of each small mesh point is 0.1 nm. AutoDock Vina was used to dock small molecules with the proteins. Finally, the dominant conformation was analyzed, and Schrodinger was used for mapping.

### 4.8. Statistical Analysis

Data analysis was performed using SPSS 19.0 (IBM Corp., Chicago, IL, USA). The expression of *PflaPBPs* was analyzed using two-way analysis of variance (ANOVA). A level of *p* < 0.05 was accepted as a significant difference.

## Figures and Tables

**Figure 1 ijms-24-00385-f001:**
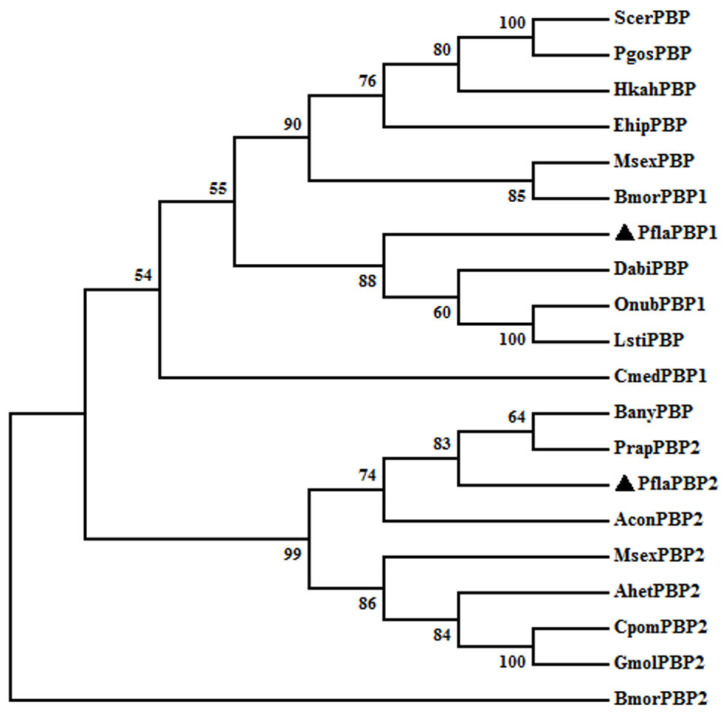
Phylogenetic tree of PflaPBP amino acid sequences with other PBPs from different insect species. The tree was constructed by the maximum likelihood method of MEGA (v7.0). Bootstrap support values (%) based on 1000 replicates are indicated. The PBP of *P. flammans* is indicated with ▲. PBPs are from different Lepidoptera. GenBank accession numbers: *P. flammans* (PflaPBP1, MK948015; PflaPBP2, MK948014), *B. mori* (BmorPBP1, AGR44778; BmorPBP2, CAL47308), *S. cerealella* (ScerPBP, AII15786), *Pectinophora gossypiella* (PgosPBP, AAF06141), *Hyposmocoma kahamanoa* (HkahPBP, XP_026321043), *Eogystia hippophaecolus* (Ehip, AOG12881), *M. sexta* (MsexPBP, XP_030025610; MsexPBP2, AAF16710), *Cnaphalocrocis medinalis* (CmedPBP1, AFG72999), *D. abietella* (DabiPBP, AZK90260), *O. nubilalis* (OnubPBP1, ADT78495), *L. sticticalis* (LstiPBP, ACD67881), *Atrijuglans hetaohei* (AhetPBP2, AKA27976), *Cydia pomonella* (CpomPBP2, AFL91693), *Grapholita molesta* (GmolPBP2, AHZ89398), *Argyresthia conjugella* (AconPBP2, AFD34183), *B. anynana* (BanyPBP, XP_023936564), *P. rapae* (PrapPBP2, XP_022118413).

**Figure 2 ijms-24-00385-f002:**
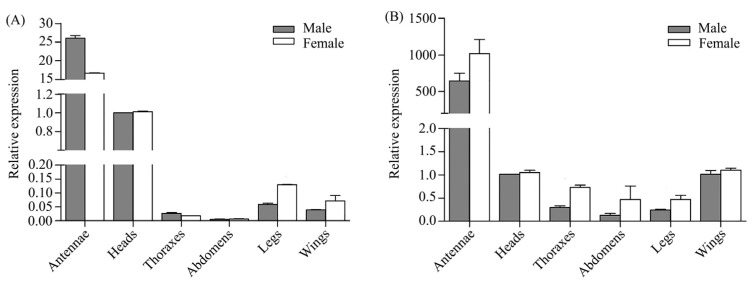
Relative transcript levels of *PflaPBP1* (**A**) and *PflaPBP2* (**B**) in different adult tissues of *P. flammans*. All values are shown as the mean ± standard error of the mean (SEM). N-values is 3.

**Figure 3 ijms-24-00385-f003:**
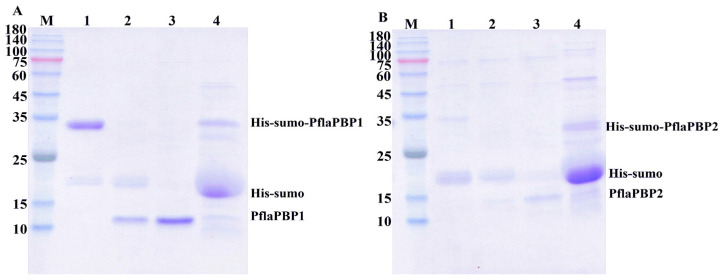
Production and purification scale-up of target proteins. SDS-PAGE with Coomassie blue staining of PflaPBP1 (**A**) and PflaPBP2 (**B**). MW, molecular weight marker. 1. Before cleavage; 2. after cleavage; 3. final sample; 4. boiled Ni resin; His-sumo, histidine-small ubiquitin related modifier.

**Figure 4 ijms-24-00385-f004:**
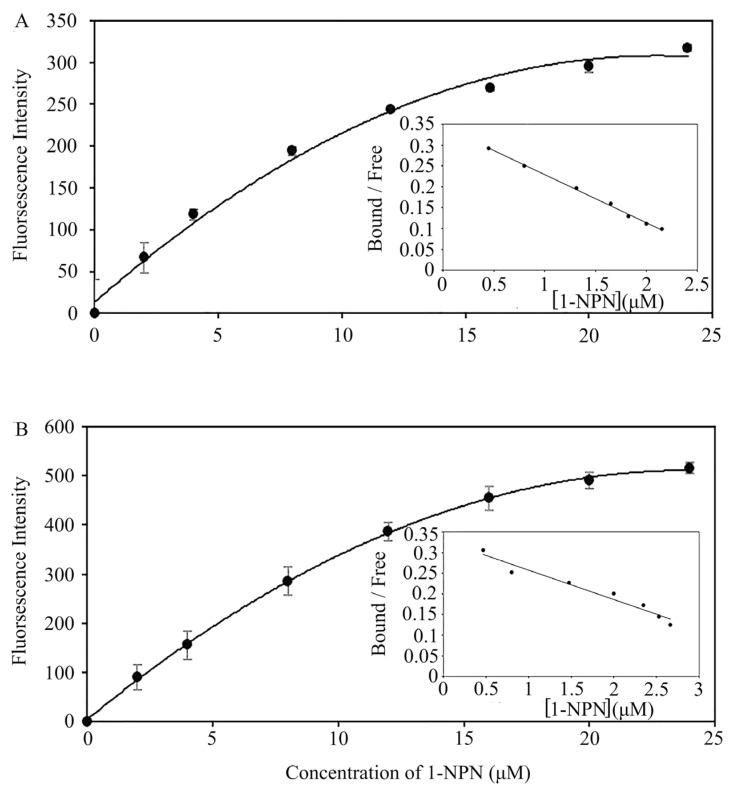
Binding curves of 1-NPN and relative Scatchard plots (insets) for PflaPBP1 ((**A**), R^2^ = 0.9597, *p* < 0.001; inset: R^2^ = 0.9971, *p* < 0.001) and PflaPBP2 ((**B**), R^2^ = 0.9833, *p* < 0.001; inset: R^2^ = 0.9585, *p* < 0.001) in P. flammans. All values are shown as the mean ± SEM. N-values is 3.

**Figure 5 ijms-24-00385-f005:**
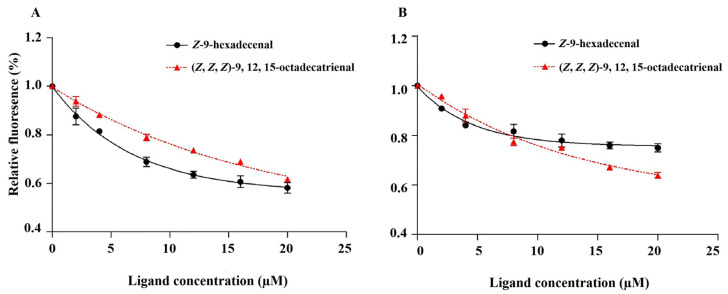
Competitive binding curves of two sex pheromones to PflaPBP1 ((**A**), Z-9-hexadecenal, R^2^ = 0.9585, *p* < 0.001; (Z, Z, Z)-9, 12, 15-octadecatrienal, R^2^ = 0.9890, *p* < 0.001) and PflaPBP2 ((**B**), Z-9-hexadecenal, R^2^ = 0.9468, *p* < 0.001; (Z, Z, Z)-9, 12, 15-octadecatrienal, R^2^ = 0.9791, *p* < 0.001) of P. flammans. All values are shown as the mean ± SEM. N-values is 3.

**Figure 6 ijms-24-00385-f006:**
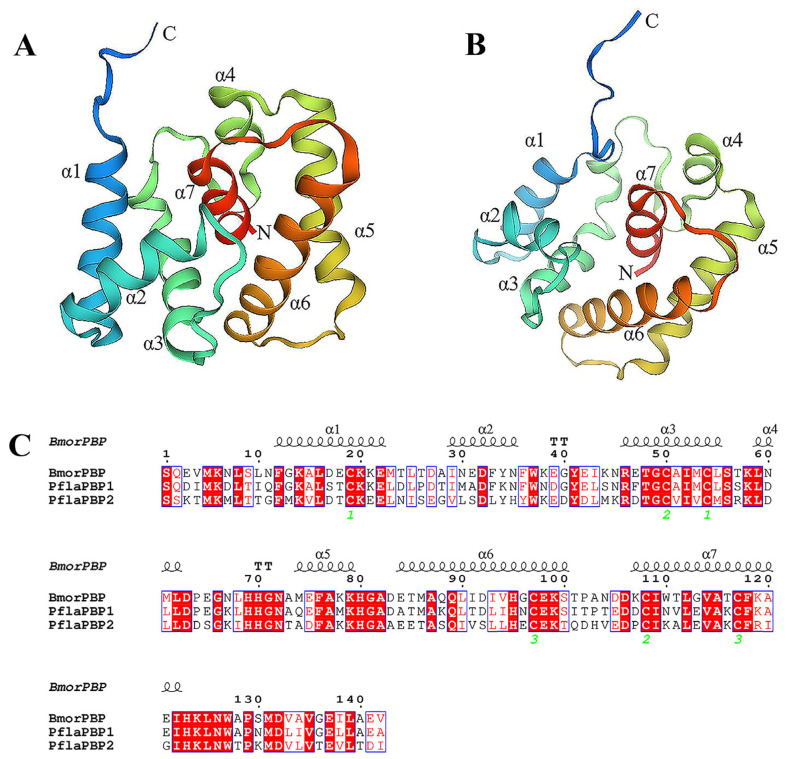
Three-dimensional structural models of PflaPBP1 (**A**) and PflaPBP2 (**B**) of *P. flammans*. The structural models were constructed by SWISS MODEL. N is the N-terminus, C is the C-terminus, and the seven helices are also labeled. (**C**) The amino acid sequence alignment of PflaPBP1, PflaPBP2, and BmorPBP.

**Figure 7 ijms-24-00385-f007:**
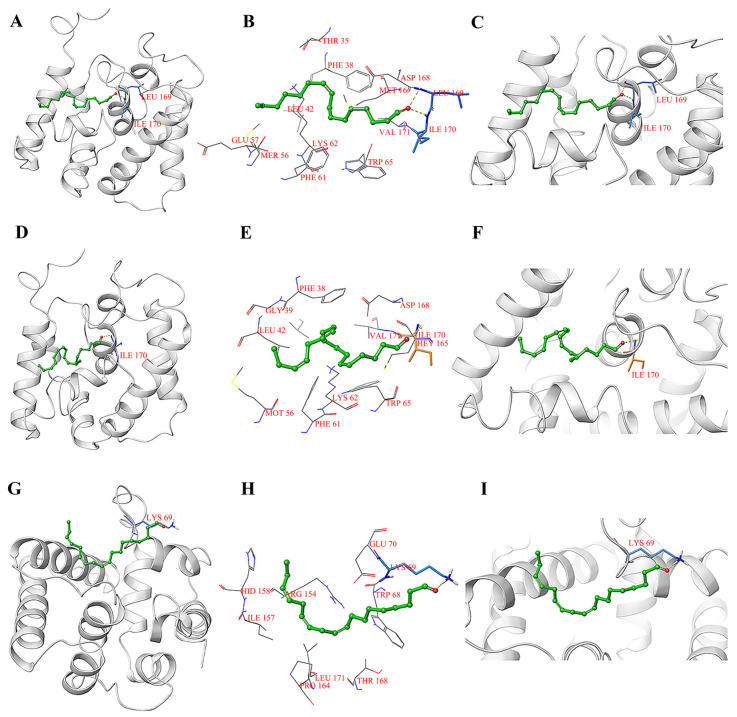
Molecular docking of PflaPBPs to two sex pheromones of *P. flammans*. (**A**) Binding mode of PflaPBP1 with *Z*-9-hexadecenal. *Z*-9-hexadecenal is presented as a green stick model with the hydroxyl oxygen in red. The blue sticks represent Leu 169 and Ile 170, which form hydrogen bonds with *Z*-9-hexadecenal alcohol; (**B**) diagram of the van der Waals interactions and hydrophobic interactions of *Z*-9-hexadecenal alcohol with PflaPBP1 key binding site residues; (**C**) the orientation and conformation of *Z*-9-hexadecenal alcohol and the hydrogen bond reaction in the PflaPBP1 active area; (**D**) binding mode of PflaPBP1 with (*Z*, *Z*, *Z*)-9, 12, 15-octadecatrienal. (*Z*, *Z*, *Z*)-9, 12, 15-octadecatrienal is presented as a green stick model with the hydroxyl oxygen in red. Blue sticks represent Ile 170 that forms hydrogen bonds with (*Z*, *Z*, *Z*)-9, 12, 15-octadecatrienal alcohol; (**E**) diagram of the van der Waals interactions and hydrophobic interactions of (*Z*, *Z*, *Z*)-9, 12, 15-octadecatrienal alcohol with PflaPBP1 key binding site residues; (**F**) the orientation and conformation of (*Z*, *Z*, *Z*)-9, 12, 15-octadecatrienal alcohol and the hydrogen bond reaction in the PflaPBP1 active area; (**G**) binding mode of PflaPBP2 with (*Z*, *Z*, *Z*)-9, 12, 15-octadecatrienal. (*Z*, *Z*, *Z*)-9, 12, 15-octadecatrienal is presented as a green stick model with the hydroxyl oxygen in red. Blue sticks represent Lys 69 that forms hydrogen bonds with (*Z*, *Z*, *Z*)-9, 12, 15-octadecatrienal alcohol; (**H**) diagram of the van der Waals interactions and hydrophobic interactions of (*Z*, *Z*, *Z*)-9, 12, 15-octadecatrienal alcohol with PflaPBP2 key binding site residues; (**I**) the orientation and conformation of (*Z*, *Z*, Z)-9, 12, 15-octadecatrienal alcohol and the hydrogen bond reaction in the PflaPBP2 active area.

**Table 1 ijms-24-00385-t001:** Effects of tissue and sex in the expression levels of *PflaPBP1* in *P*. *flammans*.

Name	Type III Sum of Squares	df	Mean Square	F	*p* Value	Partial Eta Squared
Corrected Model	2364.730 ^a^	11	214.975	1651.377	*p* < 0.001	0.999
Intercept	507.917	1	507.917	3901.664	*p* < 0.001	0.994
Tissue	2233.953	5	446.791	3432.113	*p* < 0.001	0.999
Sex	21.300	1	21.300	163.621	*p* < 0.001	0.872
Tissue × Sex	109.477	5	21.895	168.194	*p* < 0.001	0.972
Error	3.124	24	0.130			
Total	2875.771	36				
Corrected Total	2367.854	35				

^a^ R^2^ = 0.999 (adjusted R^2^ = 0.998).

**Table 2 ijms-24-00385-t002:** Effects of tissue and sex in the expression levels of *PflaPBP2* in *P*. *flammans*.

Name	Type III Sum of Squares	df	Mean Square	F	*p* Value	Partial Eta Squared
Corrected Model	3,672,802.496 ^a^	11	333,891.136	28.018	*p* < 0.001	0.928
Intercept	699,379.439	1	699,379.439	58.687	*p* < 0.001	0.710
Tissue	3,464,396.971	5	692,879.394	58.142	*p* < 0.001	0.924
Sex	34,945.539	1	34,945.539	2.932	*p* > 0.050	0.109
Tissue × Sex	173,459.986	5	34,691.997	2.911	*p* < 0.050	0.378
Error	286,009.418	24	11,917.059			
Total	4,658,191.353	36				
Corrected Total	3,958,811.915	35				

^a^ R^2^ = 0.928 (adjusted R^2^ = 0.895).

**Table 3 ijms-24-00385-t003:** Binding affinities of two sex pheromones to PflaPBPs of P. flammans.

Ligands	CAS No.	PflaPBP1		PflaPBP2	
		IC_50_ (μM)	Ki (μM)	IC_50_ (μM)	Ki (μM)
*Z*-9-hexadecenal	56219-04-6	29.13	21.26	148.92	108.68
(*Z*, *Z*, *Z*)-9, 12, 15-octadecatrienal	26537-71-3	52.73	38.48	33.95	24.80

**Table 4 ijms-24-00385-t004:** Primers for gene cloning and qRT-PCR in *P. flammans*.

Gene	Primer Sequence (5′-3′)	Amplicon Length (bp)	PCR Efficiency	Regression Coefficient
Gene cloning				
*PflaPBP1*	F: AGCAAAACTGGTACGTGAGA	740	n.a.	n.a.
	R: AAGCAAAGTGAATGCGTTTTATGT			
*PflaPBP2*	F: TGATCAGAGAGTTGAACGTGAA	600	n.a.	n.a.
	R: CCTTCATTCAATGCCTGGTCC			
qRT-PCR				
*PflaPBP1*	F: TGAAGAGGGATACTGGGTGTGT	140	1.011	0.999
	R: TTGTGAAGCGGTTTCCTCGG			
*PflaPBP2*	F: GGAAATCGG TTCACGGGTTG	100	0.994	0.999
	R: TCATTGCGAATTCCTGGGCA			
*GAPDH*	F: AACTGCCTTGCTCCACTAGC	145	0.988	0.998
	R: GAGCACCACGACCATCTCTC			
*TUB2*	F: CAACTACGCACGAGGACACT	130	1.136	0.998
	R: CACCGCCAAACGAGTGAAAC			

n.a., not applicable.

## Data Availability

Not applicable.

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
