# Peer review of "Molecular Characterization and Expression Patterns of Two Pheromone-Binding Proteins from the Diurnal Moth Phauda flammans (Walker) (Lepidoptera: Zygaenoidea: Phaudidae)"

_ijms, 2022, doi:10.3390/ijms24010385_

Round 1
Reviewer 1 Report
Although the study is, in general, well executed and well presented, I have some basic concerns about the statistical analyses.
1. The authors used a one-way ANOVA for differences between tissues and t-test for differences between sexes. This approach is not very correct and should be replaced by a two-way anova, in which the effects of tissues and sex are analysed simultaneously. With this approach, the authors will be also able to test the interaction term tissue x sex, which might be very interesting for this study. Thus, I strongly recommend replacing the one-way ANOVA + t-tests with a two-way ANOVA, in which sex is the second factor.
2. There is no indication about the liens in figures 4 and 5 were fitted. Fitting procedures should be explained. Also provide goodness -of-fit statistics and p-values if appropriate.
Minor comments
Please, check parentheses for authorities and use them according to taxonomical rules. For example, Antheraea polyphemus Cramer requires parentheses since the species was originally described in another genus. So the correct name is Antheraea polyphemus (Cramer). I suspect that most, if not all the speceis cited in this paper require parentheses.
Line 17: lepidopterous insects -> Lepidoptera
37-8: dedicated sensory organs (e.g., the antennae) and sometimes on other parts of the body [1] -> on the antennae or, more rarely, on other parts of the body [1].
41: in the antennae of male moths -> in the antennae of males
48: in antennae of female moths in many insect species, e.g., -> in antennae of female moths such as
70: add space between L. dispar and [24],
71: L. dispar[24],
71: In contrast, castniid moths -> A notable exception is represented however by castniid moths, which
74-5: To our knowledge, numerous studies -> Numerous studies
87: Please, add some references
123: I think you mean more than one PBP1. So, I suggest saying PBP1s, if it is acceptable
134-42: replace ; with ,
130-142: add authorities
148-55: This part should be rephrased according to the results of a two-way ANOVA.
I suggest something like:
The expression levels of PflaPBP1 and PflaPBP2 varied significantly among adult tissues (Tables …… - REPORT THE ANOVA TABLES, Figure 2). Specifically, expression levels [of WHAT? PflaPBP1? PflaPBP2? or both combined?] in the antennae of males and females were 241.17 times and 267.38 times the sum of other tissues, respectively. Differences in PflaPBPs expression were also found between sexes (Tables ….). There were significant (????) interactions between sex and tissues, showing that PflaPBP1 and PflaPBP2 are expressed at different levels in the same tissue in different sexes. In particular, the expression of PflaPBP1 was 1.56 times higher in the antennae of males than in those of females (Table ….), and the expression of PflaPBP2 was 1.57 times higher in the antennae of females than in those of males (Table ……).
179: The results showed that the maximum -> The maximum
192: infinity?
Figure 7: Names of amino acids in red are difficult to read because the font is too small
324: Many insect species release sex pheromones in diurnal moths of the superfamily Zygaenoidea, which can be used in the monitoring and forecasting of these moths in the field [31,34-39]. -> Many species of Zygaenoidea release sex pheromones which can be used in the monitoring and forecasting of these diurnal moths in the field [31,34-39].
336: accordance with the characteristic of male-biased expression in antennae -> accordance with the
male-biased expression in antennae
362: Second, feminine expression -> Second, female expression
395: lepidopteran insects, such as moths and butterflies (such as Spodoptera littoralis and Chrysodeixis includens) [65-71]. -> lepidopterans such as Spodoptera littoralis (Boisduval) and Chrysodeixis includens (Walker) [65-71].
400: because it is unable to collect -> because we were unable to collect [please, expalin the reason you were unable]
407: as Spodoptera frugiperda [72] -> as Spodoptera frugiperda (J. E. Smith) [72].
408: Meanwhile, it also has the possibility of -> Meanwhile, there is also the possibility of
410: The expression “are convinced of two” sounds strange to me
418: What do you mean with some reports? You report here only one study. Please, clarify.
424: Sitotroga cerealella [77]. However, -> Sitotroga cerealella (Olivier) [77]. However,
444: 26 (±1) °C, a relative humidity -> 26 (±1) °C, with a relative humidity
447: according to reported methods [78] -> following Mao et al. [78]
450: Were individuals killed before these operations? How? Please, specify.
453: But you used only three replicates in the analyses. I suggest specifying this point here.
484: lepidoptera -> lepidopteran
493: please, explain these formulas
558: B. mori in italics
Author Response
Reviewer: 1
Comments and Suggestions for Authors
Although the study is, in general, well executed and well presented, I have some basic concerns about the statistical analyses.
AQ 1: The authors used a one-way ANOVA for differences between tissues and t-test for differences between sexes. This approach is not very correct and should be replaced by a two-way anova, in which the effects of tissues and sex are analysed simultaneously. With this approach, the authors will be also able to test the interaction term tissue x sex, which might be very interesting for this study. Thus, I strongly recommend replacing the one-way ANOVA + t-tests with a two-way ANOVA, in which sex is the second factor.
Response: Thanks for your reasonable suggestion. The expressions of PflaPBPs were reanalyzed using two-way ANOVA.
AQ 2: There is no indication about the liens in figures 4 and 5 were fitted. Fitting procedures should be explained. Also provide goodness -of-fit statistics and p-values if appropriate.
Response: According to your suggestions, we added goodness-of-fit statistics and p-values in the captions of figures 4 and 5. And, the fitting procedures were also provided in the section of Materials and Methods.
Minor comments
AQ 3: Please, check parentheses for authorities and use them according to taxonomical rules. For example, Antheraea polyphemus Cramer requires parentheses since the species was originally described in another genus. So the correct name is Antheraea polyphemus (Cramer). I suspect that most, if not all the speceis cited in this paper require parentheses.
Response: We checked and corrected Latin scientific name of insects in the whole MS.
AQ 4: Line 17: lepidopterous insects -> Lepidoptera
Response: Done.
AQ 5: 37-8: dedicated sensory organs (e.g., the antennae) and sometimes on other parts of the body [1] -> on the antennae or, more rarely, on other parts of the body [1].
Response: Done.
AQ 6: 41: in the antennae of male moths -> in the antennae of males
Response: Done.
AQ 7: 48: in antennae of female moths in many insect species, e.g., -> in antennae of female moths such as
Response: Done.
AQ 8: 70: add space between L. dispar and [24],
Response: Done.
AQ 9: 71: In contrast, castniid moths -> A notable exception is represented however by castniid moths, which
Response: Done.
AQ 10: 74-5: To our knowledge, numerous studies -> Numerous studies
Response: Done.
AQ 11: 87: Please, add some references
Response: Done.
AQ 12: 123: I think you mean more than one PBP1. So, I suggest saying PBP1s, if it is acceptable
Response: Done.
AQ 13: 134-42: replace ; with ,
Response: Done.
AQ 14: 130-142: add authorities
Response: Done.
AQ 15: 148-55: This part should be rephrased according to the results of a two-way ANOVA.
I suggest something like:
The expression levels of PflaPBP1 and PflaPBP2 varied significantly among adult tissues (Tables …… - REPORT THE ANOVA TABLES, Figure 2). Specifically, expression levels [of WHAT? PflaPBP1? PflaPBP2? or both combined?] in the antennae of males and females were 241.17 times and 267.38 times the sum of other tissues, respectively. Differences in PflaPBPs expression were also found between sexes (Tables ….). There were significant (????) interactions between sex and tissues, showing that PflaPBP1 and PflaPBP2 are expressed at different levels in the same tissue in different sexes. In particular, the expression of PflaPBP1 was 1.56 times higher in the antennae of males than in those of females (Table ….), and the expression of PflaPBP2 was 1.57 times higher in the antennae of females than in those of males (Table ……).
Response: We all agree with your comment. We rephrased this paragraph according to the result from two-way ANOVA.
AQ 16: 179: The results showed that the maximum -> The maximum
Response: Done.
AQ 17: 192: infinity?
Response: This is a spelling mistake. We changed “infinity” to “affinity”.
AQ 18: Figure 7: Names of amino acids in red are difficult to read because the font is too small
Response: Font of names of amino acids in these figures was readjusted.
AQ 19: 324: Many insect species release sex pheromones in diurnal moths of the superfamily Zygaenoidea, which can be used in the monitoring and forecasting of these moths in the field [31,34-39]. -> Many species of Zygaenoidea release sex pheromones which can be used in the monitoring and forecasting of these diurnal moths in the field [31,34-39].
Response: Done.
AQ 20: 336: accordance with the characteristic of male-biased expression in antennae -> accordance with the male-biased expression in antennae
Response: Done.
AQ 21: 362: Second, feminine expression -> Second, female expression
Response: Done.
AQ 22: 395: lepidopteran insects, such as moths and butterflies (such as Spodoptera littoralis and Chrysodeixis includens) [65-71]. -> lepidopterans such as Spodoptera littoralis (Boisduval) and Chrysodeixis includens (Walker) [65-71].
Response: Done.
AQ 23: 400: because it is unable to collect -> because we were unable to collect [please, expalin the reason you were unable]
Response: According to your suggestion, we elaborated on that.
AQ 24: 407: as Spodoptera frugiperda [72] -> as Spodoptera frugiperda (J. E. Smith) [72].
Response: Done.
AQ 25: 408: Meanwhile, it also has the possibility of -> Meanwhile, there is also the possibility of
Response: Done.
AQ 26: 410: The expression “are convinced of two” sounds strange to me
Response: We corrected this description.
AQ 27: 418: What do you mean with some reports? You report here only one study. Please, clarify.
Response: We rephrased this sentence.
AQ 28: 424: Sitotroga cerealella [77]. However, -> Sitotroga cerealella (Olivier) [77]. However,
Response: Done.
AQ 29: 444: 26 (±1) °C, a relative humidity -> 26 (±1) °C, with a relative humidity
Response: Done.
AQ 30: 447: according to reported methods [78] -> following Mao et al. [78]
Response: Done.
AQ 31: 450: Were individuals killed before these operations? How? Please, specify.
Response: The individuals are living. We supplied it in the MS.
AQ 32: 453: But you used only three replicates in the analyses. I suggest specifying this point here.
Response: We corrected it.
AQ 33: 484: lepidoptera -> lepidopteran
Response: Done.
AQ 34: 493: please, explain these formulas
Response: Done.
AQ 35: 558: B. mori in italics
Response: Done.

Reviewer 2 Report
The authors present a thorough study reporting on pheromone-binding proteins from the antenna of the diurnal moth Phauda flammans. I only have some minor comments made directly on the attached PDF. In particular, the use of uppercase and lowercase letters to designate significant differences needs to be clarified in the figure captions. I recommend the manuscript be accepted after minor revision.

Author Response
Reviewer: 2
AQ 1: 157: Details need to be added to the figure caption to clarify the use of uppercase vs lowercase letters. Are the uppercase letters being used to compare among the various tissues for males, and lowercase for females?
Response: According to the suggestion of Reviewer 1, who suggest replacing the one-way ANOVA + t-tests with a two-way ANOVA. We accepted this suggestion. So, we deleted uppercase vs lowercase letters in the Figure 2.
AQ 2: 172: Since figures should stand alone from the text, His-sumo needs to be defined in the figure caption.
Response: Done.
AQ 3: 464: What criteria were used to identify candidate PBPs from the database?
Response: PBP and pheromone-binding protein were used as key words to screen the annotated chemosensory genes by the results of Nr annotation from the antenna transcriptome database. And we added explain in the MS.

Round 2
Reviewer 1 Report
The manuscript has been substantially improved.
I have a few minor comments:
1) Tables 1 and 2
Change captions as follows:
Caption to Table 1
Table 1. Effects of tissue and sex in the expression levels of PflaPBP1 in P. flammans.
Caption to Table 2
Table 2. Effects of tissue and sex in the expression levels of PflaPBP1 in P. flammans.
In the table, replace Tissues with Tissue, and Genders with Sex; use R2 instead of R Squared
2) Figure 4 is very difficult to read because of the too small size of the insets. Please, place the two panels A and B vertically, instead of horizontally, so they can be larger and readable. Also Isuggest writing the caption as follows (I assume that the “left” graphs are the main graphs, and that the “right” graphs are the insets, and that the Scatchard plots are these insets):
Figure 4. Binding curves of 1-NPN and relative Scatchard plots (insets) for PflaPBP1 (A, R2 = 0.9597, P < 0.001; inset: R2 = 0.9971, P < 0.001) and PflaPBP2 (B, R2 = 0.9833, P < 0.001; inset: R2 = 0.9585, P < 0.001) in P. flammans. All values are shown as the mean ± SEM. N-values is 3.
3) 519: into nonlinear -> with nonlinear
Author Response
Response to Reviewer: 1
AQ 1: Tables 1 and 2
Change captions as follows:
Caption to Table 1
Table 1. Effects of tissue and sex in the expression levels of PflaPBP1 in P. flammans.
Caption to Table 2
Table 2. Effects of tissue and sex in the expression levels of PflaPBP1 in P. flammans.
In the table, replace Tissues with Tissue, and Genders with Sex; use R2 instead of R Squared
Response: Done.
AQ 2: Figure 4 is very difficult to read because of the too small size of the insets. Please, place the two panels A and B vertically, instead of horizontally, so they can be larger and readable. Also Isuggest writing the caption as follows (I assume that the “left” graphs are the main graphs, and that the “right” graphs are the insets, and that the Scatchard plots are these insets):
Figure 4. Binding curves of 1-NPN and relative Scatchard plots (insets) for PflaPBP1 (A, R2 = 0.9597, P < 0.001; inset: R2 = 0.9971, P < 0.001) and PflaPBP2 (B, R2 = 0.9833, P < 0.001; inset: R2 = 0.9585, P < 0.001) in P. flammans. All values are shown as the mean ± SEM. N-values is 3.
Response: Done.
AQ 3: 519: into nonlinear -> with nonlinear
Response: Done.
